# Impact of Gender and Feature Set on Machine-Learning-Based Prediction of Lower-Limb Overuse Injuries Using a Single Trunk-Mounted Accelerometer

**DOI:** 10.3390/s22082860

**Published:** 2022-04-08

**Authors:** Sieglinde Bogaert, Jesse Davis, Sam Van Rossom, Benedicte Vanwanseele

**Affiliations:** 1Human Movements Biomechanics Research Group, Department of Movement Sciences, KU Leuven, 3001 Leuven, Belgium; sam.vanrossom@materialise.be (S.V.R.); benedicte.vanwanseele@kuleuven.be (B.V.); 2Department of Computer Science, Leuven.AI, KU Leuven, 3001 Leuven, Belgium; jesse.davis@kuleuven.be

**Keywords:** running, machine learning, lower-limb overuse injury, accelerometery

## Abstract

Even though practicing sports has great health benefits, it also entails a risk of developing overuse injuries, which can elicit a negative impact on physical, mental, and financial health. Being able to predict the risk of an overuse injury arising is of widespread interest because this may play a vital role in preventing its occurrence. In this paper, we present a machine learning model trained to predict the occurrence of a lower-limb overuse injury (LLOI). This model was trained and evaluated using data from a three-dimensional accelerometer on the lower back, collected during a Cooper test performed by 161 first-year undergraduate students of a movement science program. In this study, gender-specific models performed better than mixed-gender models. The estimated area under the receiving operating characteristic curve of the best-performing male- and female-specific models, trained according to the presented approach, was, respectively, 0.615 and 0.645. In addition, the best-performing models were achieved by combining statistical and sports-specific features. Overall, the results demonstrated that a machine learning injury prediction model is a promising, yet challenging approach.

## 1. Introduction

Physical activity is beneficial for both physical and mental health [1,2,3]. However, it also carries the risk of becoming injured, which entails associated costs and negative consequences [4]. The negative physical, psychological, and economic impact that injuries can provoke highlights the importance of injury prevention [5,6,7]. Overuse injuries account for the majority of injuries sustained during physical activities. For example, approximately 80% of running-related injuries are overuse injuries [8].

Overuse musculoskeletal injuries can result from a combination of intrinsic and extrinsic risk factors. Intrinsic factors are person-specific factors that influence the susceptibility of an individual to an injury. Instances of intrinsic factors include age, gender, body composition (e.g., weight, fat mass, and BMI), injury history, and fitness level [9,10,11,12]. Extrinsic factors are all factors externally acting on an individual that could be a contributing cause of an injury, such as training errors, excessive load, running biomechanics, fatigue, and inappropriate equipment [9,12,13]. It is the combined effect of intrinsic and extrinsic risk factors and their complex interactions that renders individuals susceptible to an injury. Some of these risk factors, such as training and running biomechanics, are modifiable and therefore have large potential for injury prevention and injury prediction [9,14]. Predicting a future overuse injury enables the performance of an intervention on modifiable factors in a timely manner to help avoid the actual development of an overuse injury. Hence, overuse injury prediction has attracted widespread interest because of its potential to help prevent an injury from occurring [15].

The nature of overuse injuries is multifactorial and the interactions between risk factors play a more important role compared to acute injuries [13]. Previous studies investigating injury risk factors disregarded the interactions between the injury risk factors or considered only a small subset of possible predefined risk factors [16,17,18,19], which could be one of the reasons for the limited success or consistency between these studies [20]. Overall, the predictive power of these current studies is limited, as they usually consider only a limited number of potential injury risk factors and focus on a linear association between a potential risk factor and an injury [21]. In general, machine learning models have higher predictive power compared to an explanatory analysis. As a result, machine learning models have attracted attention for injury prediction tasks. An injury prediction machine learning approach tries to learn automatically a model that predicts the future occurrence of an injury from a set of input features describing the data. Therefore, it is crucial for a machine learning model’s performance to have a set of features that describe informative and relevant characteristics of the data concerning injuries.

The complexity of the risk factors for overuse injuries implies the need for a complex feature representation. Crafting such a set by hand is a time-consuming and challenging task for humans relying on domain knowledge [22]. Automatic feature construction methods allow the extraction of information from acceleration time series automatically. The main disadvantage of this approach is the limited interpretability of the extracted features for the practitioner, as their construction is not driven by domain knowledge. The informativeness of automatic acceleration-derived features, also denoted as statistical features, was demonstrated by Op De Beéck et al. [23]. The automatically constructed statistical features were helpful towards the prediction of fatigue with a machine learning model. In contrast, the inclusion of sports-specific features did not improve the performance of predicting the rating of perceived exertion compared to a model using only the statistical features (i.e., ones not based on domain knowledge). However, no studies have investigated if including statistical features of the raw acceleration signal could lead to better prediction of overuse injuries. An automatic feature construction method generates a large quantity of features with potential information regarding the prediction of overuse injuries. A machine learning approach is therefore needed to identify in an automatic way which combinations of factors are important out of a large set of potential injury risk factors and surrogate indicators.

In recent years, multiple machine-learning-based injury prediction models have been proposed to tackle the challenging injury prediction task [5,15,21,24,25,26,27,28,29,30,31]. In most machine-learning-based injury prediction studies, the input data are collected by measuring a set of tests performed by the participants or monitoring the participants on regular moments. Furthermore, the current literature on machine-learning injury prediction models largely focuses on elite athletes that excel in one particular sport, e.g., running [24], soccer [15,31], or football [25,26,27]. As a result, the machine learning models and findings from these studies might not be transferable to athletes practising different sports. Moreover, Winter et al. [32] found preliminary evidence that injury-related factors of runners depend on their skill levels of a long-distance overground run, suggesting that the results of the elite-level injury prediction models might not be generalizable to athletes performing the sport on a different level. In addition, several studies focus on injuries of male athletes only [15,26,27,33]. As females and males demonstrate differences in kinematics during physical activity such as running, and previous research indicated gender specificity in factors related to an injury [19], it is important to explore if a model trained on exclusively male data is applicable to female data.

The general aim of this study is to investigate the ability of a machine learning model to predict the occurrence of an LLOI based on data collection using 3D accelerometers during a 12 min sports event in a general active population. The second objective of this study is to acquire insight into the meaningfulness of features for injury prediction. The role of the directionality and the type of features (statistical or sports-specific) in the injury-predictive capability of a machine learning model will be examined. We hypothesize that models using statistical features have higher performance than models using statistical features based on the study of Op De Beéck et al. [23]. The third objective concerns the impact of gender on injury prediction. We hypothesize that gender-specific models outperform models suitable for both genders.

## 2. Materials and Methods

### 2.1. Participants

In total, 204 first-year undergraduate students (141 males, 63 females) from the movement science program at KU Leuven in Belgium participated in the study. Data from students in first-year cohorts in two academic years, 2019–2020 and 2020–2021, were collected. Each participant engaged in this study on a fully voluntary basis. There were no positive or negative consequences for the students of the academic program associated with the willingness or refusal to participate. Because some students (partly) retake their first year, it was ensured that no participants were recruited twice. Before the measurements, all participants underwent medical screening and gave written consent in accordance with the Declaration of Helsinki. In addition, the ethics committee of Gasthuisberg, University Hospital Leuven, approved the study.

The academic program of the participants included 10 h a week of physical exercise for 26 weeks per academic year. The sports practised during these physical exercise classes were dance, track and field, gymnastics, swimming, basketball, handball, soccer, and volleyball. In addition, most participants participated in sports outside the program.

As a requirement of the academic program, all participants had to consult a sports physician at the Sports Medical Advice Center (SMAC) of University Hospital Leuven in the event of a (suspected) injury. After six months, the physicians communicated whether an injury diagnosis was established for each participant, and if so, what the diagnosis was. The physicians established an injury diagnosis when (1) a reduction in the amount of physical activity was recommended, and (2) medical advice or treatment was needed [18]. Subsequently, the established injuries were classified as an LLOI or a non-LLOI injury. An LLOI is defined as an injury on the lower limbs that is the consequence of a musculoskeletal load exceeding the musculoskeletal capacity. The onset is gradual and the symptoms of the injury are progressive [34]. When an injury diagnosis was established, but not consistent with the above definition of LLOI (e.g., due to an onset matched to a single traumatic event, or an injury to an upper limb and trunk), the injury was classified as a non-LLOI injury.

Participants satisfying at least one of the following exclusion criteria were eliminated from the study. (1) The first exclusion criterion was the diagnosis of a non-LLOI injury. It was assumed that a non-LLOI injury influenced the probability of developing an LLOI as a non-LLOI injury might hinder the performance of physical activity. (2) Participants with an unknown injury status were excluded. These participants probably dropped out of the program. However, due to privacy rules, this could not be verified. (3) Participants with missing values for one of the features were excluded from this study. Missing values were the consequence of an incomplete questionnaire. (4) A fourth exclusion criterion was the requirement of at least 10 min of running out of the 12 min. If there occurred more than two minutes of non-running stages during the test, either there were problems with the IMU or the subject rested too much during the test, leading to a different level of fatigue compared to the other participants.

### 2.2. Data Collection

At the start of the academic year, all participants performed a Cooper test, which entails running at a steady pace for as far as possible within 12 min. The test took place, after a warm-up session, on an outdoor synthetic 400 m track. After the test, the total distance covered during 12 min was recorded.

For each cohort, the measurement took place on two separate days, with four to six different sessions on each day. At the start of each session, an inertial measurement unit (IMU) in a custom-designed silicone pocket was positioned and tightly secured with a belt on the lower back, over the L3 to L5 spinal segments. The position of this sensor was based on the results from a study by Schütte et al. [35]. The IMU (Byteflies, Antwerp, Belgium) was used to measure tri-axial acceleration with a sampling range of 1000 Hz, a 16-bit resolution, and a measuring range from −16 to +16 times the gravitational acceleration.

In addition to the running test, each participant filled in a questionnaire including questions regarding their weight, height, gender, previous injuries, and whether they wore insoles.

### 2.3. Data Preprocessing

The data preprocessing started by selecting the part of the acceleration signal that corresponded to the Cooper test. The period of standing still after the warm-up and before the Cooper test began enabled us to easily identify the appropriate part of the data. Figure 1 provides a schematic overview of how the raw acceleration data of the Cooper test were preprocessed prior to computing features. To account for the tilt of the runner and ensure that the analysis focused on the dynamic component of the acceleration arising from the runner, a tilt correction procedure proposed by Moe-Nilssen [36] was implemented. This procedure was applied to the entire time series as the first step of the preprocessing procedure. The tilt correction procedure assumes a constant tilt. Although this assumption was anticipated to be violated over the entire trial, it was expected that the results would be accurate enough to localize the running stages, which were defined as the portions of the test where the participant was running (i.e., not walking). Non-running stages could occur due to walking or a disturbance in the acceleration measurement. Since there was a clear difference between running and walking in the acceleration signal, running stages were detected by analysis of the peaks of the vertical acceleration of the tilt-corrected signal. This localization information (red dashed line in Figure 1) was used to extract running stages from the non-tilt-corrected data. Subsequently, the first minute and last ten seconds of every running stage were removed to ensure that the signals corresponded to steady-state running. The subjects performed the tests in groups of approximately 25, where all subjects started the test simultaneously. Therefore, it took each subject a small amount of time at the start to find their own place and pace. The last ten seconds were removed as some participants either accelerated to improve the covered distance or decelerated as they approached the end of the Cooper test or ended a running stage. The resulting cropped time series were corrected for tilt and static gravity in non-overlapping windows of two minutes. The smaller window size compared to step one minimized the violation of the constant tilt assumption of the tilt correction procedure. The obtained acceleration signals were suitable for feature extraction.

### 2.4. Feature Construction

From the preprocessed time series, two sets of features were calculated: statistical features and sports-specific features. The statistical features were extracted automatically from the time series and were selected irrespective of the context. The sports-specific features were all features that were suggested and selected by domain experts for this specific injury prediction setting.

The set of statistical features was calculated using the publicly available Python package TSFuse, an automated feature construction system [22,37]. The setting fast was used, and more details of the constructed features can be found in the article by De Brabandere et al. [22]. The sports-specific features referred to surrogate measures for dynamic stability, dynamic loading, and spatio-temporal measures. Computing some sports-specific features first required the performance of step detection, which was done using the automated step detection procedure proposed by Benson et al. [38]. For this study, the set of sports-specific features included the root-mean-square ratio, step regularity, stride regularity, sample entropy, impact, standard deviation of sample entropy, standard deviation of impact, angle in medial–lateral direction during the first two minutes and the last two minutes, angle in anterior–posterior direction during the first two minutes and the last two minutes, and step time. This set of features was calculated for the resultant signal and the signals in the medial–lateral, vertical, and anterior–posterior directions. The selection of these features was motivated by a study that established a link between a part of these features on the one hand, and fatigue and LLOI on the other hand [18].

Once all the features were calculated, a subject-based min-max normalization was performed. The minimum and maximum values required for this normalization procedure were calculated from the first six ten-second windows per subject and per feature [23]. The medial–lateral and anterior–posterior angles were excluded from the min-max normalization and only a centering was performed. The angle value of the first two-minute window was subtracted from the angle value during the last two-minute window.

The time-series-calculated features were supplemented with the distance covered during the Cooper test and features obtained from the questionnaire, which included gender, weight, height, previous injuries, and whether the participant wore insoles. Any feature that had the same value for all subjects was removed because such features have no discriminative value. In total, 75 statistical features were removed. Table 1 gives the number of remaining features per category.

### 2.5. Model Construction

The machine learning approach of LLOI prediction adopted in this study consisted of several steps, executed within nested cross-validation (CV). A nested CV approach was adopted as, in general, it allows the estimation of the method’s performance with a lower bias compared to a k-fold CV [39,40,41]. As Figure 2 visualizes, an internal three-fold CV procedure was run on the training data to select an appropriate set of hyperparameters. Subsequently, the best configuration was evaluated on the held-aside test set of the fold.

In each fold of the internal CV, four steps were performed on the training data to train the method. The first step was one-hot encoding of the categorical features and standardization of the numerical features. The standardization took place per feature over all subjects. This is in contrast to the earlier performed min-max normalization, which was subject-based and hence was performed per subject and per feature. In the second step, a principal component analysis (PCA) reduced the dimensionality of the feature space. Subsequently, an ANOVA F-value feature selection approach (the scikit-learn function *SelectKBest* [42]) selected a subset of the obtained principal components. The last step was the training of a machine learning model that aimed to predict the occurrence of an LLOI based on the selected principal components. Three machine learning algorithms were considered, specifically L1-regularized logistic regression (LR), random forest trees (RFT), and support vector machines (SVM). These algorithms are commonly employed for injury prediction modeling [15,21,25,27,30,43,44]. For these models, Figure 3 shows a simplified diagram of the classification process of new or test instances. A grid search was employed that repeated all four steps for each combination of hyperparameter values. The best-performing hyperparameter setting, according to the internal CV, was selected. The full training set was used to learn a model for this setting, which was then applied to the test set to estimate the generalization ability. By repeating this entire procedure of Figure 2 for each fold, the generalization performance of the method with a grid search hyperparameter tuning could be estimated.

The following hyperparameters were tuned: the number of PCA components, the number of selected PCA features, and the hyperparameters of the model. For all models, the possible options for the number of PCA components were 10,15,20,25, and 30 while the number of selected PCA features ranged from two to six. For L1-regularized logistic regression, the options for the solver were liblinear and saga. The range for the inverse of regularization strength was case-dependent, as different datasets were used. Likewise, the range of the inverse regularization strength for support vector machines was case-dependent. All support vector machines did share the options of kernel (*polynomial*, *radial basis function*, or *sigmoid*) and the possible range of degree (1, 2, or 3). For random forest trees, 100 estimators were used; the maximum depth of each tree was 3 or 4, the minimum number of samples required to split an internal node was 3 or 4, the split criterion was *gini* or *entropy*, and the minimum impurity decrease was case-specific as different datasets were used.

This study used the area under the receiver operating characteristic curve (AUC) for hyperparameter tuning and model evaluation. In addition, the Brier score is reported. Platt scaling was used to derive probabilities for the support vector machine [45].

To deal with the imbalance in the classes LLOI versus non-LLOI, weights were given to the classes [46]. The assigned weight of a class is inversely proportional to the frequency of the class. Models were trained on the entire dataset or a gender-specific subset of the data. Because of the smaller amount of female data, compared to male-specific or mixed-gender data, a six-fold CV (with a 3-fold internal CV) was implemented and the option of 30 PCA components was omitted for the female-specific models. Furthermore, models trained on different subsets of the features (sports-specific set, statistical set, or their combination) will be considered. The features derived from the questionnaire are included in both sets.

### 2.6. Importance Values of Features

Since a logistic regression model is interpretable, the importance value of different features can be calculated and compared. The importance value of a feature indicates how the natural logarithm of the logit transform $\left(logit\left(p\right)=ln\left[\frac{p}{1-p}\right]\right)$ of the probability of an LLOI occurrence *p* changes in response to a change in the feature. The importance value $importance_{f}$ of a feature *f* was calculated as follows:(1)$$\begin{array}{cc} \; & importance_{f}=abs\left(\sum_{i}^{k}w_{i}PC_{if}\right)\\ \; & \mathrm{where}\;PC_{i}=\sum_{j}^{m}PC_{ij}f_{j}. \end{array}$$
with *k* being the number of predictors of the logistic model, $w_{i}$ the coefficient of the logistic model for predictor *i*, $PC_{i}$ a principal component that is the *i*-th predictor of the logistic regression model, $PC_{if}$ the coefficient of associated with feature *f* in the decomposition of $PC_{i}$, and *m* the number of features. In Equation (Equation 1), the absolute value of the sum is taken since the size of the influence is of interest and not the sign. Therefore, there is no distinction made between overall positive and negative influences on logit(p).

Because of the large number of features, the importance values were analyzed per category. There were three analyses performed, each time considering a different categorization criterion to subdivide the entire set of features. The first category division criterion is the directionality of the features. Features have been derived from the acceleration signal in the medial–lateral, anterior–posterior, vertical, and resultant directions. The non-directional category comprises all features that do not describe characteristics of the acceleration signal in a specific direction. The non-directional features include the questionnaire-derived features and step time (a sports-specific feature). The second division criterion was based on the type of feature, i.e., sports-specific, statistical, or questionnaire-based. For the third division criterion, the features are subdivided on both the directionality and type of the features—for instance, all sport-specific features derived from the acceleration signal in the vertical form for one category.

The importance values of all features belonging to the same category are combined by taking an average. This operation accounts for the imbalance in the number of features in each category. In addition, the importance values of the categories are expressed as a proportion. Hence, the relative average importance of category *c* is calculated as
(2)$$importance_{c}^{category}=\frac{\frac{\sum_{i}^{k_{c}}importance_{i}}{k_{c}}}{\sum_{j}^{n}\left(\frac{\sum_{i}^{k_{j}}importance_{i}}{k_{j}}\right)}$$
with $k_{c}$ being the number of features belonging to category *c*, *n* the number of categories considered in the analysis, and $importance_{i}$ the importance value of features *i* as defined in Equation (Equation 1).

The described data preprocessing, feature construction, pipeline procedure, and importance calculation were performed in Python 3.9.7.

## 3. Results

Of the 204 subjects, 100 subjects participated in September 2019 and 104 participated in 2020. There were five subjects with a total running time of fewer than 10 min. In total, 27 participents experienced a non-LLOI injury. Twenty-five of the non-LLOI injuries were a consequence of a traumatic event: 17 ankle distortion or inversion injuries, three injuries on the upper limbs, two fibula fractures, two foot injuries (traumatic injury at calcaneus or peroneus), and one knee injury. In addition, two overuse injuries at the upper limbs or lower back were observed. In addition, the injury status of six participants was unknown and five participants had a missing value for at least one of the features. Hence, these 43 participants were excluded from the analysis. There were 161 subjects left for model training and evaluation (109 males and 52 females), of which 41 subjects had an LLOI. Table 2 shows the types and the incidence of LLOI among the participants during the six-month follow-up period. Table 3 summarizes the descriptive characteristics of the participants included in the analysis. The male participants had a mean body mass of 71.38(±7.56) kilogram and a mean height of 179.86(±6.33) centimeters. For the female participants, a mean mass of 62.51(±6.63) kilogram and a mean height of 167.19(±5.99) centimeters was observed.

### 3.1. Model Performance

Table 4 summarizes the mean test scores for the all-data models, which are applicable to both considered genders. These results estimate the generalization performance of the algorithm with grid-search-CV hyperparameter tuning. To indicate the uncertainty in this performance estimate, the standard deviation of the AUC values is provided. The highest mean AUC score (0.557±0.091) was obtained with a support vector machine (SVM) that used the entire set of features. The average Brier score of this model was 0.193±0.022. Since the AUC is only slightly greater than 0.5, the classification performance is only slightly better than random guessing.

Gender-specific models have been trained and evaluated on a gender-specific subset of the data. Table 5 reports the performance estimates for models trained only on data from female subjects. The best-performing model in terms of AUC score is a logistic regression (LR) model using the entire set of features. This model achieves a mean AUC score of 0.645±0.056 and a mean Brier score of 0.190±0.021. The second place is for the models using solely the statistical features. The lowest-performing models in terms of AUC are the models that only use sports-specific features.

Table 6 reports the performance of the models trained only on data from male subjects. In this setting, a logistic regression model outperformed all other models in terms of AUC. This model obtained a mean AUC of 0.615±0.063.

Comparing the mean Brier score obtained for the different ML algorithms reveals that the support vector machine models attained the lowest and thus the best results. This holds for the female-specific, male-specific, and all-data models, irrespective of the (sub)set of features employed.

Figure 4 shows a pooled receiver operating characteristic (ROC) curve for the best-performing models for the mixed-gender and gender-specific models, i.e., an all-feature logistic regression model for the male- and female-specific models and an all-feature support vector machine model for the mixed-gender model. The ROC curves are constructed based on the combined test-set predictions of the five or six models generated during the cross-validation approach.

### 3.2. Importance of Feature Categories

The best-performing gender-specific models are logistic regression models and therefore have an interpretable character. Relative average importance values have been calculated to provide insight on the importance of each category of feature on logit(p). Figure 5 shows these relative average importance values. One plot is shown for each division criterion described in Section 2.6, i.e., the directionality of the feature, the type of feature, and the combination of directionality and type of the features. Note that the range of the vertical axes differs for the three plots.

Figure 5a compares the relative average importance of the features, subdivided according to the direction of the acceleration signal used to derive the features. An extra category (non-directional) is provided to cover the features that are not associated with a specific direction. For the male model, changes in the features derived from the vertical acceleration signal are, on average, most influential on logit(p). For the female model, features derived from the medial–lateral direction are most influential. For both genders, the anterior–posterior-directed features come in second place, and the non-directional features are least influential on logit(p). Overall, Figure 5a shows that, except for the female non-directional category, the differences between the relative importance of feature categories of different directions are only limited.

Figure 5b shows the relative average importance for sports-specific, statistical, and questionnaire features. For the male-specific case, the importance of the sports-specific and statistical features is similar. In the female-specific model, the statistical features are, on average, more important compared to the sports-specific features. For both genders, the questionnaire-based features have the lowest average importance.

Figure 5c visualizes the relative average importance for categories based on both directionality and the type of the feature. This shows that, irrespective of gender, the sports-specific features derived from an acceleration signal in the horizontal plane are more influential compared to the non-directional or vertical sports-specific features. Furthermore, the highest relative average importance for the male- and female-specific model is, respectively, the statistical features in the vertical direction and the statistical features in the medial–lateral direction.

## 4. Discussion

Injury prediction remains a challenging task. In this study, we trained a model with the purpose of predicting the occurrence of an LLOI up to six months after data collection. The available information consisted of features extracted from time series of tri-axial accelerations collected during a single Cooper test, supplemented with participants’ intrinsic characteristics from a questionnaire. The best-performing model with an AUC of 0.645 is gender-specific, including statistical as well as sports-specific features.

### 4.1. Role of Gender Subdivision on Model

Gender-specific models delivered the best results regarding our injury prediction goal. The highest mean AUC scores observed for the female- and male-specific models are, respectively, 0.645±0.056 and 0.615±0.063. These results are in line with previously reported injury prediction models using machine learning techniques, where the AUC score ranged between 0.52 and 0.87 [5]. However, compared to the study by López-Valenciano et al. [33], our AUC results are considerably lower as they obtained an AUC of 0.747 for a machine learning model predicting lower-extremity muscle injuries in 132 male professional soccer and handball players. This model takes as input a 120 min testing session assessing the individual characteristics, physiological measures, and neurological measures of the athlete. The testing in the current study is less elaborate and requires only 10% of the testing duration of the study of López-Valenciano et al. [33]. Furthermore, less homogeneity in skill level is expected for our participants compared to a group of professionals. Moreover, the injury-related factors might be different for different skill levels [32], which might explain the inferior results of our study. The results obtained in our study approach closely the AUC performance of the models presented in a paper by Jauhiainen and coworkers [21]. They reported an AUC of 0.65 for predicting knee and ankle injuries in 314 young basketball and football players based on data from a set of physical tests in a laboratory setting.

The performance of the all-data models is inadequate regarding the injury prediction objective of this study. The mean AUC score of the best-AUC-performing models is 0.542±0.075. The all-data models perform only marginally better compared to a random guesser. The superior estimated performance of gender-specific injury prediction models is in accordance with previous research that demonstrated gender specificity in injury-related factors [19,32]. Gender-specific models allow us to account for differences in injury-related factors to a greater extent compared to the all-data models where the PCA features incorporate gender.

### 4.2. Sports-Specific and Statistical Features

To analyze the injury-predictive informative capability and complementarity of sports-specific features and statistical features, different models have been trained, each exploiting a different set of features. Overall, the model utilizing all available features outperformed the corresponding models using only a subset of the features. Only the all-data random forest tree model is an exception to this. Furthermore, for the majority of the results, models using the statistical features scored better compared to the corresponding models with the sports-specific features. This indicates that, in terms of an LLOI prediction, the combination of sports-specific and statistical features is more informative than either set of features alone. Hence, the statistical features and sports-specific features are partly complementary. Furthermore, the results denote that the statistical features are more informative than the sports-specific features. Previous studies mainly utilized a combination of intrinsic and sports-specific features to predict the injuries in machine learning. Although the features are extracted from a different kind of data, as no other injury prediction study was found that used a single Cooper test for input data, it is expected that the models presented in the literature could benefit from the inclusion of statistical features.

The male-specific models based on sports-specific features achieve higher performance for AUC compared to either the female-specific or the all-data models. There are two possible explanations for this result. A first possible explanation is that the sports-specific features are more explanatory regarding injury prediction for males compared to females. Alternatively, the larger sample size of the male dataset compared to the female sample set could also explain this observation. Despite the larger male dataset, the mean AUC score for the models exploiting the combination of statistical features and sports-specific features is consistently lower for male-specific models compared to female-specific models. This observation could suggest that the statistical and sports-specific features are more complementary for females compared to males.

Although the interpretability of the statistical features is limited, they do improve the practicality of the models by improving the predictive performance. The prediction model as a whole can be used as guidance for indicating subjects at risk for developing an LLOI. This knowledge allows a focus on preventive measures for the susceptible subjects. Nevertheless, further research is recommended to be able to translate the statistical features into interventions that could improve the value of these statistical features.

### 4.3. Importance of Feature Categories

The importance of sports-specific and statistical features is similar in the male-specific model, while, in the female-specific models, the statistical features are clearly more important. This is in line with the observation of the performance of each of the gender-specific models using only the subset of the features.

The non-directional features are, especially for the female-specific model, of low relative importance concerning LLOI prediction. Since the non-directional features are mainly composed of questionnaire-based features, this observation suggests that monitoring acceleration during a fatiguing run is crucial for good performance in the prediction of LLOIs.

Looking at the combination of type and directionality of the features, the sports-specific features derived from the acceleration in the horizontal plane are most influential, among the sports-specific features, with respect to the prediction of LLOI for either gender. Previous research has demonstrated that running-induced fatigue mainly affects the variability of the sports-specific features derived from the acceleration signals of the horizontal plane [47]. As a result, one of the possible speculations is that response to fatigue is indicative for LLOI prediction.

It should be noted that the specific feature importance values are solely indicative and should be interpreted with care. Each feature is considered separately for the calculation of its importance value under the assumption that all other feature values are constant. However, this is not a realistic case. Some features are related and changing one of them results in a change in the other one (e.g., weight and BMI, or acceleration in the medial–lateral direction and the resultant acceleration). Moreover, even if features are not related, a change in only one of the features while keeping the other features constant might be unrealistic.

### 4.4. Strengths and Limitations

The obtained results (mean AUC of 0.645 and 0.615) are comparable to the performance of models described in the literature of injury prediction models [21,27]. In this study, a nested CV approach was used, in contrast to most studies, which use a *k*-fold CV. As a result, it is presumed that the presented results are a less biased estimate of the true error. Furthermore, our models only require information of a single Cooper test measured by a single tri-axial accelerometer. As there is no need for regular testing or a laboratory setting, the practicality of this model is high. Moreover, the study is characterized by a relatively large number of participants. In addition, no or only a small minority are elite athletes, which potentially makes the model more applicable to the wider public.

We are aware that our research has several limitations. Subjects with missing values for at least one of the features or with a non-LLOI injury were excluded from the analysis. Although missing values for features are believed not to be missing completely at random, it was assumed that the bias introduced by removing the subjects from the study is minimal. The same assumption was supposed for the removal of subjects with a non-LLOI injury. All the participants were physically active, of similar age (approximately 18 years old), and undergoing approximately the same training load due to the academic program. As a consequence, the generalization capability of these results and models to a wider population is limited. Although we have similar results to previous studies [5,21,27,43], the models are not high-performing and could be improved. More risk factors, such as previous sports participation, anthropometric characteristics, running shoes, etc., could be included in the model and be a potential step towards enhancing the performance. In addition, as the measurements were performed at the beginning of a six-month investigation period, which coincided with the start of the academic program, some information might be missing. It is expected that follow-up measurements could provide complementary information and, as a result, have the potential to improve performance. Future studies on the topic of injury prediction with machine learning are recommended to be concerned with aspects such as the transferability of the model or approach to new settings, or the expansion of the feature set to cover more intrinsic and extrinsic injury-related factors.

## 5. Conclusions

The purpose of the present study was to examine the ability of a machine learning model to predict the occurrence of an LLOI. The outcomes from this study make two main contributions to the current literature. The first main finding to emerge from this study is that the model’s performance considerably improved by splitting the data according to gender and training gender-specific models. The generalization performance, measured by a mean AUC score, of the best-performing method, fitted with a grid-search CV hyperparameter tuning, was 0.645 and 0.615 for, respectively, female- and male-specific models. A second finding of this research is that the statistical and sports-specific features are partly complementary as the combination of both sets resulted in the best-performing models. On average, models using only statistical features deserve a second place, while models with sports-specific features come in last. Additionally, it was observed that the medial–lateral-acceleration-derived features were the most influential feature category in the female logistic regression model. For the male model, the features derived from the vertical acceleration came out on top.

In recent years, machine learning techniques have demonstrated potential regarding the growing field of sports injury prediction. The findings of the presented study may contribute to this field and be valuable for future studies as the results demonstrate the importance of considering gender specificity and a suitable feature set.

## Figures and Tables

**Figure 1 sensors-22-02860-f001:**
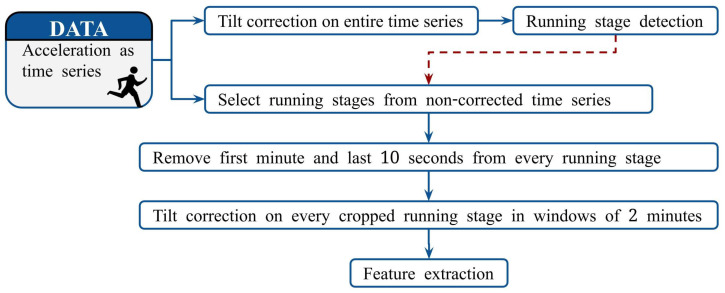
Diagram of the preprocessing procedure of the data. A blue arrow indicates the flow of data. The red dashed arrow denotes the flow of information about the location of the running stages within the entire time series.

**Figure 2 sensors-22-02860-f002:**
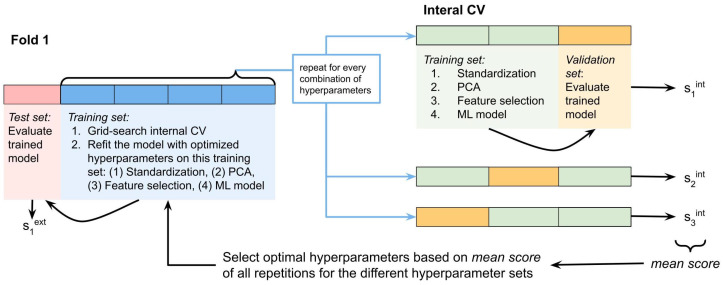
Visualization of a part of the model construction approach.

**Figure 3 sensors-22-02860-f003:**
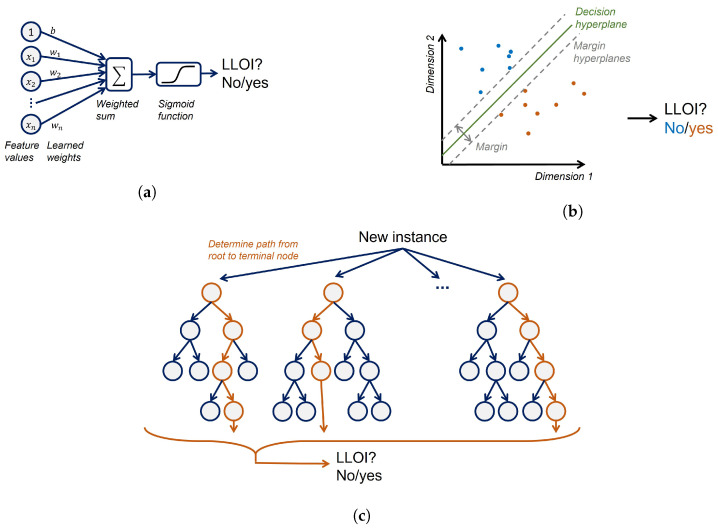
Simplified diagrams showing how the three used models classify instances. (**a**) In a logistic regression model, a weighted sum of the feature values is mapped with a sigmoid function onto the probability of a future occurrence of a lower-limb overuse injury (LLOI) (**b**) A support vector machine learns a maximum margin hyperplane that separates LLOI from non LLOI. New instances are classified by determining on what side of the learned hyperplane they lie. (**c**) A random forest consists of multiple trees. For a new instance, a path from the root to a terminal node (orange path) is determined according to the instance’s feature values. Each terminal node contains a probability distribution over the classes, and the final prediction is determined by averaging these probability distributions.

**Figure 4 sensors-22-02860-f004:**
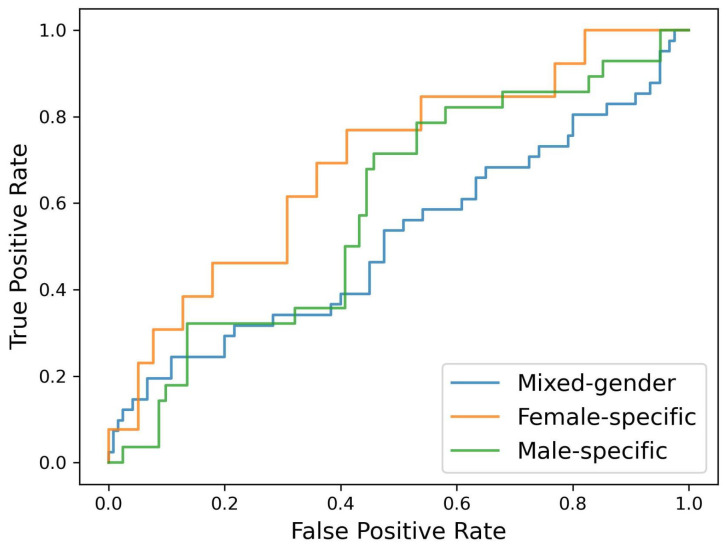
Pooled ROC curve for the best-performing mixed-gender, male-specific, and female-specific model.

**Figure 5 sensors-22-02860-f005:**
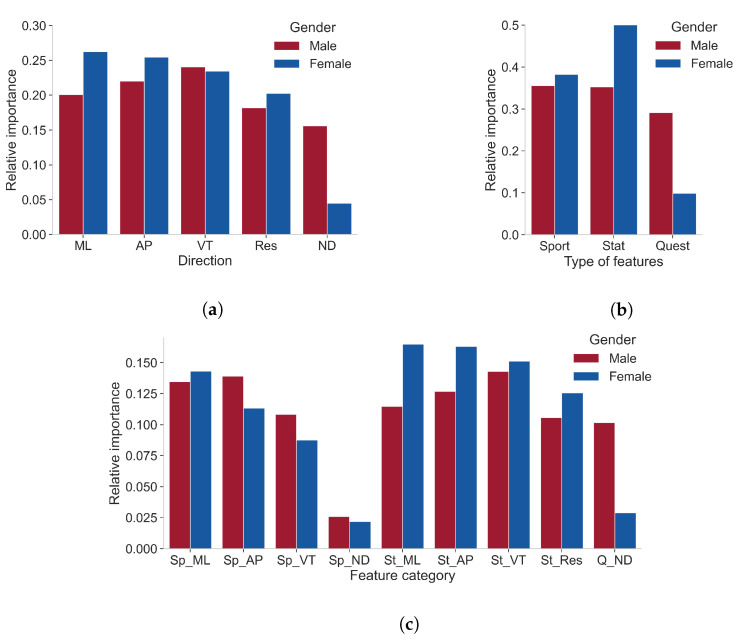
Relative average importance of several feature categories for the male- and female-specific logistic regression model. Two different criteria for subdividing the features into categories are considered: (**a**) directionality of feature, i.e., medial–lateral (ML), anterior–posterior (AP), vertical (VT), and resultant direction (Res), or non-directional (ND), (**b**) the type of feature, i.e., sports-specific (Sport), statistical (Stat), or questionnaire-based (Quest), and (**c**) both the type of feature (before underscore; Sports specific (Sp); Statistical (St); Questionnaire (Q)) and the directionality of the feature (after underscore).

**Table 1 sensors-22-02860-t001:** Number of features per category.

Feature Category	Number of Features
All	312
Sports-specific	24
Statistical	281
Questionnaire	7

In this table, the categories of sports-specific and statistical features do not include the features extracted from the questionnaire. When discussing the performance of the models using sports-specific features or statistical features, the questionnaire features are included.

**Table 2 sensors-22-02860-t002:** Type and incidence of lower-limb overuse injuries diagnosed during the six-month follow-up period.

Type of Injury	Number of Participants
Medial Tibial Stress Syndrome	26
Muscle overuse injury	5
Knee or hip overuse injury	4
Patellofemoral pain	3
Iliotibial band syndrome	2
Bone overuse	1

**Table 3 sensors-22-02860-t003:** Descriptive characteristics of all participants included in the analysis. Values of continuous variables are expressed as mean ± standard deviation. Discrete variables are expressed as the number of participants with the characteristic.

		Male		Female	
Characteristics	All	LLOI	No Injury	LLOI	No Injury
Number	161	26	81	13	39
Mass (kg)	68.52 ± 8.37	70.85 ± 6.35	71.57 ± 7.93	62.86 ± 7.65	62.40 ± 6.25
Height (cm)	175.8 ± 8.6	180.4 ± 5.5	179.7 ± 6.6	166.6 ± 6.8	167.3 ± 5.7
Distance test (m)	2746 ± 431	2916 ± 485	2948 ± 315	2254 ± 195	2373 ± 233
Previously injured	61	14	25	7	15
Insoles	22	6	11	2	3

**Table 4 sensors-22-02860-t004:** Results for the models trained using only the entire dataset for different sets of features and algorithms. Either all features (All), only the sports-specific features (Sport), or only the statistical features (Stat) are used. The reported values are a mean across the scores obtained for each of the five CV folds.

			Mean CV Results	
	Features	Model	AUC (±Std)	Brier Score
1	All	LR	0.526 (±0.144)	0.249 (±0.010)
2	All	RFT	0.512 (±0.036)	0.248 (±0.002)
**3**	**All**	**SVM**	**0.557** (±0.091)	0.193 (±0.022)
4	Sport	LR	0.475 (±0.109)	0.254 (±0.003)
5	Sport	RFT	0.483 (±0.033)	0.248 (±0.001)
6	Sport	SVM	0.492 (±0.092)	0.196 (±0.005)
7	Stat	LR	0.453 (±0.067)	0.257 (±0.007)
8	Stat	RFT	0.518 (±0.036)	0.246 (±0.003)
9	Stat	SVM	0.512 (±0.096)	0.191 (±0.008)

LR: logistic regression; RFT: random forest tree; SVM: support vector machine; std: standard deviation. The model with the highest AUC score across all models is printed in bold.

**Table 5 sensors-22-02860-t005:** Results for the models trained using only the data from females for different sets of features and algorithms. Either all features (All), only the sports-specific features (Sport), or only the statistical features (Stat) are used. The reported values are a mean across the scores obtained for each of the six CV folds.

			Mean CV Results	
	Features	Model	AUC (±Std)	Brier Score
**1**	**All**	**LR**	**0.645** (±0.056)	0.190 (±0.028)
2	All	RFT	0.520 (±0.134)	0.244 (±0.004)
3	All	SVM	0.560 (±0.105)	0.185 (±0.028)
4	Sport	LR	0.464 (±0.097)	0.256 (±0.010)
5	Sport	RFT	0.502 (±0.100)	0.246 (±0.006)
6	Sport	SVM	0.415 (±0.158)	0.220 (±0.062)
7	Stat	LR	0.645 (±0.110)	0.229 (±0.066)
8	Stat	RFT	0.518 (±0.040)	0.246 (±0.002)
9	Stat	SVM	0.603 (±0.057)	0.190 (±0.021)

LR: logistic regression; RFT: random forest tree; SVM: support vector machine; std: standard deviation. The model with the highest AUC score across all models is printed in bold.

**Table 6 sensors-22-02860-t006:** Results for the models trained using only the data from males for different sets of features and algorithms. Either all features (All), only the sports-specific features (Sport), or only the statistical features (Stat) are used. The reported values are a mean across the scores obtained for each of the five CV folds.

			Mean CV Results	
	Features	Model	AUC (±Std)	Brier Score
**1**	**All**	**LR**	**0.615** (±0.063)	0.245 (±0.005)
2	All	RFT	0.533 (±0.067)	0.247 (±0.001)
3	All	SVM	0.531 (±0.047)	0.197 (±0.027)
4	Sport	LR	0.495 (±0.064)	0.253 (±0.003)
5	Sport	RFT	0.475 (±0.050)	0.247 (±0.001)
6	Sport	SVM	0.451 (±0.031)	0.190 (±0.009)
7	Stat	LR	0.576 (±0.061)	0.241 (±0.007)
8	Stat	RFT	0.485 (±0.029)	0.248 (±0.002)
9	Stat	SVM	0.509 (±0.076)	0.205 (±0.028)

LR: logistic regression; RFT: random forest tree; SVM: support vector machine; std: standard deviation. The model with the highest AUC score across all models is printed in bold.

## Data Availability

The data presented in this study are available on request from the corresponding author.

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
