# Peer review of "Impact of Gender and Feature Set on Machine-Learning-Based Prediction of Lower-Limb Overuse Injuries Using a Single Trunk-Mounted Accelerometer"

_sensors, 2022, doi:10.3390/s22082860_

Round 1

Reviewer 1 Report

  1. General Comments

The author conducted a cohort study to assess the occurrence of a lower-limb overuse injury (LLOI) after six months from baseline. The participants were the undergraduate students in the university, and the author recorded the acceleration time series data with using a single tri-axial accelerometer during Cooper test and received the questionnaire. The machine learning algorithms were used for prediction of the occurrence of LLOI at six months period, and the contribution of feature values to the prediction model was estimated. The performance of each model considered the gender difference was important for prediction of future LLOI, and the directions of acceleration signal which should be considered for prediction were different between the gender. These findings suggest a potential of accurate prediction of occurrence of LLOI with using simple assessments in future. However, I have some concerns that should be addressed regarding the study design, methodology, and interpretation of results.

  1. Specific comments

Major

  • i ) First of all, it is recommended to add detailed explanations for the exclusion criteria. In this study, the author excluded the participants with a non-LLOI injury which is “an injury not occurring on the lower limbs or when the injury is acute”. However, the details were not be written. Moreover, the author stated “Thirty-three participants experienced a non-LLOI injury” in line 271. It seemed a little too big. It is recommended to add the detail of non-LLOI, the reason and the validity of this occurrence rate. Also, they stated about the “missing data” at limitation session, but the bad-quality data was not fully explained. The reason why the participants were excluded is important for accurate apprehension for the study result.
  • ii ) The description about the inclusion criteria is also considered to be insufficiently. In this study, first-year undergraduate students from the movement science program at the university were recruited. The participation to this study should not be included in the academic program, but it seems that the measurement of this study is a part of academic program. The author obtained the written consent from all participants, but it is unknown that there was no relationship between the participate to this study and the academic program result. To avoid any misunderstanding, it is recommended to add the description about there was certain opportunity to refuse to participate or the description about there were no relationship between the academic program and study result in this study.
  • iii ) In this study, the true label is considered as the occurrence of LLOI, but the criteria of LLOI is seemed to be subjective and that the description about LLOI was not sufficient. There were 2 criteria for LLOI; (1) a reduction in the amount of physical activity was recommended, or (2) medical advice or treatment was needed. These descriptions are considered subjective. If there were medical verification for diagnosis as LLOI, it is recommended to added such description. Moreover, what kind of LLOI were occurred when participants consult to a sports medicine physician was not sufficiently described. The author referenced the previous report, but found only similar description. The referenced study included the description about LLOI such as injury types, and the author may be the same research group, but the participants were not same. It is recommended to add the information about what kind of LLOI injury was occurred in this study.
  • iv ) There were some unclear points about the interpretation about the importance values. The author stated “the sports-specific features derived from the acceleration in the horizontal plane are most influential, among the sports-specific features, with respect to the prediction of LLOI”. It seems that this description is the explanation about female-specific model. However, the highest values about the direction in Figure 3 and appendix A1 for male were vertical. I recommend to add an explanation about male model.

Minor

    • i ) The author stated in the limitation session about missing values, but there is no explanation why the data were missing and when the data were missing.
    • ii ) It is unclear that the reason why the first minute and last ten seconds were thought not to be steady-state running.
    • iii ) The author stated “The sports-specific features refer to surrogate measures for some extrinsic factors of an overuse injury” in line166. The sports-specific features (included “root-mean-square ratio, step regularity, stride regularity…”) did not necessarily appear to be external factors. I recommend to add further explanation or change the words.
    • iv ) In the result session, the author compared the relative importance values in each direction and each type of features. The certain criteria or meaningful difference are unclear.
    • v ) Although the author refers to athletes with respect to the application of this model in line 426, the participants were limited to the undergraduate students in the university. Moreover, the age of participants was not described and might be limited.
    • vi ) The inclusion of statistical features seems to have benefit as the author stated in lines377-380. However, it seems difficult to prevent for the LLOI from the result of statistical features in clinical setting. How to utilize these results from statistical features for prevention of LLOI is unclear. I recommend to add explanation about this point.
    • vii ) The reason to decide the AUC from this study as “moderate” is unclear.
    • viii ) The reference should be added about the words “previous studies” in line 436.
    • ix ) In the conclusion session, I recommend to add a description about the values of this study for future work.

Reviewer 2 Report

The manuscript evaluated the impact of gender and features on lower-limb overuse injuries by some machine learning (ML) models and an accelerometer. The approach is interesting; however, many major problems need to be addressed in this research.

  1. Please improve English.
  2. Page 4 “Running stages were detected…. of the tilt-corrected signal.” Can you ensure that this is a good process? How about the other motions such as jumping, sprinting, etc.?
  3. Section Model construction, please provide the pictures for three models: LR, RFT, SVM.
  4. Page 6, maybe the paragraph which explained ROC and AUC is unnecessary.
  5. Section Importance values, how about the importance of these values for other models: RFT and SVM?
  6. The results of ML models should show in pictures of ROC/AUC.
  7. How about the accuracy and confusion matrix of three ML models?
  8. What is the size of the training set and the testing set? If it is 161, I don’t think this size is enough for an ML model.
  9. Besides, there is only one appendix picture; it should be put inside the main text.
  10. The AUC value (~ 0.6) is a poor value for an ML model. This value shows that the model can not be used in practical applications. Maybe the too small dataset led to poor results. Therefore, the analysis of the features based on those poor results is not enough for reliability.
  11. Small issue: spelling error at a keyword.

I think this approach is interesting. However, the paper lacks a good dataset. The low accuracy is not good to evaluate the models or can be used in practical applications.

Round 2

Reviewer 1 Report

The authors well addressed my point. Ready to be accepted.

Reviewer 2 Report

Thanks for the revision.
In Model construction, please provide some simple diagrams for three algorithms. These pictures will support the readers who are not in the ML field.
